# Silicon Improves the Redox Homeostasis to Alleviate Glyphosate Toxicity in Tomato Plants—Are Nanomaterials Relevant?

**DOI:** 10.3390/antiox10081320

**Published:** 2021-08-23

**Authors:** Cristiano Soares, Pedro Nadais, Bruno Sousa, Edgar Pinto, Isabel M. P. L. V. O. Ferreira, Ruth Pereira, Fernanda Fidalgo

**Affiliations:** 1GreenUPorto—Sustainable Agrifood Production Research Centre and INOV4AGRO, Biology Department, Rua do Campo Alegre s/n, Faculty of Sciences of University of Porto (FCUP), 4169-007 Porto, Portugal; up201604645@fc.up.pt (P.N.); bruno.sousa@fc.up.pt (B.S.); ruth.pereira@fc.up.pt (R.P.); 2LAQV/REQUIMTE, Laboratory of Bromatology and Hydrology, Department of Chemical Sciences, Faculty of Pharmacy, University of Porto (FFUP), Rua de Jorge Viterbo Ferreira no. 228, 4050-313 Porto, Portugal; ecp@ess.ipp.pt (E.P.); isabel.ferreira@ff.up.pt (I.M.P.L.V.O.F.); 3School of Health, Department of Environmental Health, P.Porto (ESS-P.Porto), Rua Dr. António Bernardino de Almeida, 400, 4200-072 Porto, Portugal

**Keywords:** herbicides, nanoparticles, stress alleviation, oxidative stress, antioxidants

## Abstract

Given the widespread use of glyphosate (GLY), this agrochemical is becoming a source of contamination in agricultural soils, affecting non-target plants. Therefore, sustainable strategies to increase crop tolerance to GLY are needed. From this perspective and recalling silicon (Si)’s role in alleviating different abiotic stresses, the main goal of this study was to assess if the foliar application of Si, either as bulk or nano forms, is capable of enhancing *Solanum lycopersicum* L. tolerance to GLY (10 mg kg^−1^). After 28 day(s), GLY-treated plants exhibited growth-related disorders in both shoots and roots, accompanied by an overproduction of superoxide anion (O_2_^•−^) and malondialdehyde (MDA) in shoots. Although plants solely exposed to GLY have activated non-enzymatic antioxidant mechanisms (proline, ascorbate and glutathione), a generalized inhibition of the antioxidant enzymes was found, suggesting the occurrence of great redox disturbances. In response to Si or nano-SiO_2_ co-application, most of GLY phytotoxic effects on growth were prevented, accompanied with a better ROS removal, especially by an upregulation of the main antioxidant enzymes, including superoxide dismutase (SOD), catalase (CAT) and ascorbate peroxidase (APX). Overall, results pointed towards the potential of both sources of Si to reduce GLY-induced oxidative stress, without major differences between their efficacy.

## 1. Introduction

“It took 200,000 years for our human population to reach 1 billion—and only 200 years to reach 7 billion” (https://mahb.stanford.edu/blog/human-population-time/, accessed on 20 July 2021). The message is clear. Human population is on the rise and, therefore, more food must be produced with fewer resources and less available land [1]. For this reason, and especially since the last half of the 20th century, agriculture is progressively more dependent on agrochemicals to achieve high yield rates. Consequently, the pesticide industry has been continuously growing over recent decades [2]. By definition, agrochemicals are chemical agents used to protect crops from diseases and pests, and/or to enhance plant growth under adverse conditions [3]. According to a recent report of the Environmental Protection Agency (EPA) of the United States, among all kinds of pesticides, herbicides account for almost 50% of the total expenditures between 2008 and 2012 worldwide [4]. 

Specifically focusing on this class, glyphosate (GLY)-based herbicides are the most sold formulations and are expected to remain as the leading chemical option for weed control in the following years (https://www.marketsandmarkets.com/Market-Reports/herbicides-357.html, accessed on 12 June 2021). Concretely in Europe, GLY use was recently renewed until the end of 2022 (https://ec.europa.eu/food/plant/pesticides/glyphosate_en, accessed on 12 June 2021). Although at the beginning, GLY [N-(phosphonomethyl) glycine] use was restricted to some areas, given its non-selective action, the development of GLY-resistant crops (e.g., maize, soybean, cotton) has largely contributed to the substantial increase in its commercialization and widespread use [5,6]. In general terms, GLY is classified as a foliar, broad-spectrum, post-emergent and systemic herbicide, acting by blocking the biosynthesis of essential amino acids, such as tryptophan, tyrosine and phenylalanine [7]. Once absorbed by the plant, GLY tends to accumulate in metabolically active sites, mainly in apical meristems, where it inhibits the action of 5-enolpyruvylshiquimato-3-phosphate synthase (EPSPS; EC 2.5.1.19). As a consequence, the shikimate pathway is compromised, resulting in an overaccumulation of shikimate and a deficit of chorismate in plant cells, ultimately inhibiting the synthesis of aromatic amino acids [7].

Since the shikimate pathway is exclusively found in plants and some species of microorganisms [8], GLY was—and still is—considered as one of the most innocuous chemical options for weed control [9]. However, especially in the last few years, concerns have been raised regarding the possible toxicity of GLY, not only for animals and humans, but also for soil organisms and non-target plants [5,6,10,11,12]. Although it is claimed that, once in contact to the soil, GLY is quickly degraded by the action of microorganisms and/or adsorbed to soil particles (DT50 of around 20 days; http://sitem.herts.ac.uk/aeru/iupac/Reports/373.htm, accessed on 20 June 2021), recent findings suggest that GLY can persist in the environment, accumulating in soils and/or being leached to surface waters [5]. For this reason, the scientific community has been gathering efforts to unravel the potential hazards of GLY to different trophic levels, from producers to consumers and decomposers (recently reviewed by Van Bruggen et al. [5]). Up to now, although there is no consensus regarding the real toxicity of GLY to animals, multiple studies have found that soil residues of GLY can impair plant growth, possibly inflicting losses in global agronomic yields. For instance, lab-scaled experiments revealed that GLY soil contamination greatly affects the growth of different crop plants, including tomato (*Solanum lycopersicum* L.), barley (*Hordeum vulgare* L.) and pea (*Pisum sativum* L.), contributing to the disruption of the redox homeostasis and imposing a severe oxidative stress condition e.g., [10,11,13]. In this way, bearing in mind that GLY use is still approved in the European Union (EU), it is of special importance to develop new ways to enhance the tolerance of non-target plants to this herbicide.

Silicon (Si) is the second most abundant element on Earth crust, being considered as a beneficial element for plant growth [14]. Although there is no consensus about its role as an essential nutrient, the involvement of Si in several metabolic pathways and physiological events is well described in the literature [15,16,17], especially regarding its ability to improve plant stress tolerance [18]. Si, applied either by soil amendment, foliar spray or seed priming, is highly recognized for its potential to reduce the negative effects of different stressful conditions on plants, acting at different levels of plant physiology, reducing the overproduction of reactive oxygen species (ROS) and boosting the plant antioxidant system [17]. Nowadays, not only bulk forms of Si are considered as promising tools to increase plant resilience, but also their nano-sized counterparts, namely silicon dioxide nanomaterials (nano-SiO_2_), are being pointed as a more efficient way to provide Si [19,20]. However, to the best of our knowledge, the effects of both Si and nano-SiO_2_ on the alleviation of GLY-induced stress are yet to be uncovered. Therefore, this work aims at exploring the beneficial effects of the application of Si, in its bulk and nano forms, on GLY-induced oxidative stress in tomato plants (*S. lycopersicum* cv. Micro-Tom). For this purpose, a set of biometrical, ecophysiological and biochemical approaches were implemented to unravel the potential of Si and nano-SiO_2_ to mitigate the stress induced by GLY, in the prevention of oxidative damage and in the efficiency of the antioxidant system.

## 2. Materials and Methods

### 2.1. Chemicals and Artificial Substrate

Sodium metasilicate pentahydrate (Na_2_SiO_3_·5H_2_O) and silicon dioxide nanomaterial (nano-SiO_2_) (hydrophilic with a particle size of 7–14 nm, a specific surface area of 200 m^2^ g^−1^ and a 99.8% purity) were purchased from Merck© and Nanostructured & Amorphous Materials Inc. (Houston, TX, USA), respectively, as powders. The characterization of nano-SiO_2_, in terms of size and shape, was previously performed by our group [21]. GLY was acquired in the form of RoundUp^®^ Ultramax (Bayer, Portugal), which is a commercial formulation containing 360 g L^−1^ GLY as potassium salt. The plant growth substrate was an artificial soil (pH 6.0 ± 0.5), containing 5% (*m/m*) organic matter, provided as peat, prepared according to the guidelines of an OECD protocol [22].

### 2.2. Plant Material and Growth Conditions

Seeds of *Solanum lycopersicum* L. cv. Micro-tom, obtained from FCUP’s seed collection (Porto, Portugal), were used as biological material in the present work. Before sowing, seeds were surface disinfected with 70% (*v/v*) ethanol and 20% (*v/v*) sodium hypochlorite (5% active chloride), supplemented with 0.05% (*m/v*) Tween-20, for 5 min each, and subsequently washed several times with deionized water (dH_2_O). Then, seeds were placed in Petri dishes containing half-strength MS medium [23] solidified with 0.625% (*m/v*) agar, and left for germination in a growth chamber, under controlled conditions of temperature (25 °C), photoperiod (16 h light/8 h dark) and photosynthetic active radiation (120 µmol m^−2^ s^−1^). After 8 days, plantlets were transferred to plastic pots (200 g OECD substrate, contaminated, or not, by 10 mg kg^−1^ GLY). To ensure nutrient availability, 120 mL of modified Hoagland solution [24] were added to a cup placed under each pot at the beginning of the assay. The communication between the cup and the pot was achieved by a cotton rope. Afterwards, dH_2_O was added when required, and plants were grown for 28 day(s) in a growth chamber, as described above.

### 2.3. Experimental Design

In order to investigate the possible ameliorating role of Si nutrition on GLY-induced toxicity in *S. lycopersicum*, plants were divided into different experimental groups (Figure 1): CTL—control plants grown in OECD substrate (negative control);Si—plants grown in OECD substrate and treated once a week with 1 mM Si by foliar spraying;Nano-SiO_2_—plants grown in OECD substrate and treated once a week with 1 mM nano-SiO_2_ by foliar spraying;GLY—plants grown in OECD substrate contaminated by 10 mg kg^−1^ GLY (positive control);GLY + Si—plants grown in OECD substrate contaminated by 10 mg kg^−1^ GLY and treated once a week with 1 mM Si by foliar spraying;GLY + nano-SiO_2_—plants grown in OECD substrate contaminated by 10 mg kg^−1^ GLY and treated once a week with 1 mM nano-SiO_2_ by foliar spraying

For each experimental condition, eight experimental replicates were considered, each one with five plants. The selection of Si, provided as Na_2_SiO_3_·5H_2_O, and nano-SiO_2_ concentrations were based on previous bibliographic records [25,26,27,28] and set as 1 mM of Si. Plants from the CTL and GLY experimental groups were sprayed weekly with dH_2_O only. After 28 day(s) of growth, individuals from four replicates, randomly selected, were collected, separated into shoots and roots, and immediately used for biometric analysis and O_2_^•−^ content; in parallel, shoots and roots of plants from the other four replicates were frozen in liquid nitrogen (N_2_) and stored at −80 °C for later use. For all studied parameters, including all biochemical procedures, samples from at least three experimental replicates were used.

### 2.4. Biometric Determinations

At the end of the assay, i.e., after 28 day(s) of growth, plants were used for the measurement of root and shoot length and biomass production. Upon separation of roots and shoots, the organ elongation was measured and, then, using a precision balance (KERN©; EWJ 300-3; KERN & SOHN GmbH, Balingen, Germany), the fresh biomass of both organs was registered.

### 2.5. Assessment of Lipid Peroxidation (LP)

LP was evaluated in terms of malondialdehyde (MDA) content, using frozen samples of roots and shoots, according to Heath and Packer [29]. After homogenization and centrifugation, extracts were mixed with 0.5% (*m/v*) thiobarbituric acid (TBA) in 20% (*m/v*) trichloroacetic acid (TCA). Following 30 min at 95 °C, the absorbance of each sample was read at 532 and 600 nm. To avoid unspecific turbidity, the obtained values at 600 nm were subtracted to those at 532 nm, and the MDA content was calculated using an ε of 155 mM^−1^ cm^−1^ and expressed as nmol g^−1^ fresh mass (fm).

### 2.6. Determination of ROS Levels—Superoxide Anion (O_2_^•−^) and Hydrogen Peroxide (H_2_O_2_)

Cellular levels of O_2_^•−^ were determined in samples of fresh roots and shoots, by incubating pieces of plant material (ca. 1 cm^2^; 200 mg), for 1 h, in 3 mL of a reaction mixture [10 mM sodium phosphate buffer (pH 7.8), 10 mM sodium azide (NaN_3_) and 0.05% (*m/v*) nitroblue-tetrazolium (NBT)] [30]. At the end, the absorbance (Abs) was registered at 580 nm. The levels of O_2_^•−^ were expressed as Abs_580 nm_ h^−1^ g^−1^ fm. Regarding H_2_O_2_, its content was evaluated following the protocol of Alexieva et al. [31], in which the extract reacts with potassium iodide (KI) to form a yellowish complex that can be measured at 390 nm. Levels of H_2_O_2_ were determined by a linear calibration curve, and expressed in nmol g^−1^ fm.

### 2.7. Quantification of Non-Enzymatic Antioxidants—Proline (Pro), Glutathione (GSH) and Ascorbate (AsA)

Pro was quantified in frozen plant samples by the ninhydrin-based colorimetric assay [32], by measuring the absorbance at 520 nm. Its levels were determined after obtaining a linear calibration curve with solutions of known concentration, and results were expressed in mg g^−1^ fm. The quantification of GSH was accomplished by following the procedure described in Soares et al. [10], in which GSH reduces 5,5′-dithiobis-(2-nitrobenzoic acid) (DTNB) to 2-nitrobenzoic acid (TNB), a reaction that can be measured at 412 nm. GSH levels were estimated from a linear calibration curve prepared with known concentrations of this antioxidant. Results are expressed on an fm basis. Ascorbate content, as well as its reduced (AsA) and oxidized (dehydroascorbate—DHA) forms, were quantified by spectrophotometry at 525 nm, based on the 2,2′-bipyridyl method [33]. Levels were estimated using a linear calibration curve obtained with AsA standards, and results expressed in µmol g^−1^ fm.

### 2.8. Extraction and Quantification of Antioxidant Enzymes—Superoxide Dismutase (SOD; EC 1.15.1.1), Catalase (CAT; EC 1.11.1.6), Ascorbate Peroxidase (APX; EC 1.11.1.11), Glutathione Reductase (GR; EC 1.8.1.7), and Dehydroascorbate Reductase (DHAR; EC 1.8.5.1)

The extraction of the main antioxidant enzymes was performed as previously described [21]. After centrifugation, the supernatant was collected and used for both, protein content quantification [34] and enzyme activity assays. In the case of SOD, an aliquot of SN was complexed with 10 µM NaN_3_. Total SOD activity was performed based on the inhibition of the photoreduction of NBT, by spectrophotometry at 560 nm [35]. For each sample, an appropriate volume of extract (30 μg of protein) was added to a reaction mixture containing 100 mM potassium phosphate buffer (pH 7.8), 0.093 mM EDTA, 12.05 mM l-methionine, 0.0695 mM NBT and 0.0067 mM riboflavin in a final volume of 3 mL. The enzymatic reaction was started by adding the riboflavin to the tubes, which were immediately placed under six fluorescent 8 W lamps for 10 min. After this period, the light source was removed in order to stop the reaction. Enzyme activity was expressed as units SOD mg^−1^ protein, in which one unit represents the amount of SOD required to inhibit NBT photoreduction by 50%. The evaluation of CAT and APX activity was accomplished by enzyme kinetics, by measuring the decomposition of H_2_O_2_ (ε_240 nm_ = 39.4 M^−1^ cm^−1^) [36] and of AsA (ε_290 nm_ = 2.8 mM^−1^ cm^−1^) [37] over 2 min. In both cases, the reaction was started by the addition of H_2_O_2_. Regarding DHAR and GR, changes in Abs at 265 and 340 nm were monitored to follow AsA (ε_265 nm_ = 14 mM^−1^ cm^−1^) production and NADPH consumption (ε_340 nm_ = 6.22 mM^−1^ cm^−1^), respectively. Results are expressed as µmol min^−1^ mg^−1^ protein. The original protocol was adapted to UV microplates, based on the optimization of Murshed et al. [38].

### 2.9. Quantification of GLY and Aminomethylphosphonic Acid (AMPA) Accumulation in Plant Tissues

The extraction of GLY and AMPA from tomato tissues was carried out according to the AOAC Official method 2000.05. Briefly, 100 mg of freeze-dried homogenised tissues (shoots or roots) were extracted with 5 mL of ultrapure water by shaking for 10 min on an end-over-end shaker. Afterward, samples were centrifugated at 10,000 rpm for 10 min (at 4 °C) and the supernatant recovered. Sample derivatization and analysis was performed according to Pinto et al. [39], with some modifications: 1 mL of the supernatant (SN) was diluted with 1 mL of internal standard (200 µg L^−1^ of GLY 1,2-^13^C_2_ ^15^N and 200 µg L^−1^ of ^13^C,^15^N-AMPA), to which 120 µL of 1% (*m/v*) NH_4_OH solution and 120 µL of FMOC-Cl (12,000 mg L^−1^ in acetone) were added. The tubes were shaken for a few seconds and incubated for 30 min at room temperature. The reaction was stopped by adding 10 μL of 6 M HCl. The derivatized extracts were filtered through a 0.45 μm PTFE filters into LC vials. GLY and AMPA were determined by liquid chromatography with tandem mass spectrometry (LC-MS/MS) using the internal standard method.

The LC-MS/MS system comprised a Waters 2695 XE separation module (Milford, MA, USA) interfaced to a triple quadrupole mass spectrometer (Quattro micro™ API triple quadrupole, Waters Micromass, Manchester, UK). The LC separation was performed using a Kinetex^®^ EVO C18 core-shell column (2.6 µm; 100 × 2.1 mm) at a flow rate of 225 µL/min. A binary gradient was used, which consisted of solvent A (10 mM ammonium bicarbonate) and solvent B (methanol). The percentage of organic modifier (B) was changed linearly as follows: 0–0.5 min, 5%; 0.5–5.5 min, 90%; 5.5–6.5 min, 90%; 6.5–6.7 min, 5%; 6.7–14 min, 5%. The injection volume was 20 µL and the column temperature was kept at 40 °C. The mass spectrometry parameters were as follows: ion mode, positive; capillary voltage, 3.00 kV; source temperature, 130 °C; desolvation temperature, 450 °C; desolvation gas flow, 600 L/h and multiplier, 650 V. High purity nitrogen (>99.999%) and argon (>99.999%) were used as the cone and collision gases, respectively. The precursor and product ions as well as the cone voltage and collision energy for each GLY-FMOC, AMPA-FMOC and ILIS-FMOC were determined by flow injection analysis and the multiple reaction monitoring (MRM) transitions, cone voltages and collision energies are listed in Table 1. Data acquisition was performed by the MassLynx V.4.1 software (Waters© Corportation, Milford, MA, USA). Results are expressed on a dry mass (dm) basis.

### 2.10. Statistical Analyses

All biometric and biochemical determinations were performed, at least, in three independent replicates (*n* ≥ 3), and results are expressed as mean ± standard deviation (STDEV). Differences between experimental groups were tested by one-way ANOVA, assuming a significance level (*α*) of 0.05. In case of significant differences, Tukey’s *post hoc* tests were performed to discriminate differences between groups. Prior to the ANOVAs, data were checked for normality and homogeneity through Shapiro–Wilk and Brown–Forsythe tests, respectively. All statistical procedures were performed in Prism 8 (GrahPad^®^, San Diego, CA, USA).

## 3. Results

### 3.1. Biometric and Growth-Related Parameters

As can be seen in Figure 2, the exposure of *S. lycopersicum* to 10 mg kg^−1^ GLY caused a marked reduction in plant development, significantly impairing the growth of both roots and shoots. This finding was effectively demonstrated when root length [F (5, 13) = 70.90; *p* < 0.05] and fresh biomass of both organs [shoots: F (5, 15) = 5.93; *p* < 0.05; roots: F (5, 15) = 46.76; *p* < 0.05] were evaluated. Inhibitions of around 50 and 70% were recorded for shoot and root growth, respectively, in comparison with the CTL (Figure 3). When GLY was not added to the substrate, the foliar spray with both sources of Si, especially nano-SiO_2_, positively affected plant growth, significantly increasing the root length (by 114%) and fresh biomass (by 27%) in comparison with the CTL (Figure 2 and Figure 3). Moreover, GLY phytotoxicity was significantly reduced by the foliar application of both Si or nano-SiO_2_ (Figure 2 and Figure 3). Indeed, plants from GLY + Si and GLY + nano-SiO_2_ groups showed a better ability to grow, reaching values closer to the CTL for root length and shoot fresh weight. Moreover, the marked reduction of root biomass in response to GLY (73% lower) was significantly counteracted by Si or nano-SiO_2_ treatments, with reduced inhibition values over the control (GLY + Si—48% reduction; GLY + nano-SiO_2_—40% reduction).

### 3.2. Lipid Peroxidation (LP)—MDA content

The MDA content, indicative of LP, showed a distinct behaviour between shoots and roots of tomato plants (Figure 4a,d). Significant differences between treatments were found for both organs [shoots: F (5, 12) = 8.04; *p* < 0.05; roots: F (5, 13) = 6.74; *p* < 0.05]. In shoots, GLY induced a pronounced increase in MDA levels, with a rise of 32% in relation to the CTL, this effect being significantly counteracted by the application of Si or nano-SiO_2_ to levels similar to the CTL. As evidenced in Figure 4d, in general, roots of GLY-exposed plants exhibited a lower LP degree (up to 41%). Despite this pattern not being changed in Si co-treated plants, co-exposure of GLY and nano-SiO_2_ resulted in MDA levels identical to the CTL in roots.

### 3.3. ROS Homeostasis—O_2_^•−^ and H_2_O_2_ Content

The production of O_2_^•−^ was significantly changed in shoots [F (5, 23) = 4.25; *p* < 0.05] and roots [F (5, 20) = 12.88; *p* < 0.05] (Figure 4b,e). The levels of O_2_^•−^ were higher in plants exposed to GLY alone, with values exceeding by 75 and 80% those found in shoots and roots of CTL plants, respectively. However, when plants were treated with Si or nano-SiO_2_, the levels of this ROS were kept identical to the CTL, especially in roots, where each treatment resulted in a decrease of around 60% compared to GLY-exposed plants (Figure 4b,e).

Concerning H_2_O_2_, no significant changes were found in shoots among all experimental groups [F (5, 13) = 2.45; *p* > 0.05; (Figure 4c)]. In roots, significant changes between treatments were found [F (5, 13) = 22.69; *p* < 0.05]. However, no statistical differences were detected between GLY and CTL plants (Figure 4f). Under the co-exposure scenario, both Si treatments contributed to decreasing the accumulation of H_2_O_2_, especially bulk Si, where a reduction of around 60% in H_2_O_2_ content was found in relation to the CTL and GLY plant groups. This was observed despite both Si materials per se having significantly increased the levels of this ROS (44% and 54%, bulk and nano, respectively), when applied alone.

### 3.4. Non-Enzymatic Antioxidants—Pro, GSH and AsA

Pro levels, illustrated in Figure 5a,d, were changed in response to GLY [shoots: F (5, 14) = 15.44; *p* < 0.05; roots: F (5, 18) = 12.14; *p* < 0.05], with values around 5.3- and 2.0-fold of those of the CTL, in both shoots and roots, respectively. Upon co-treatment with Si or nano-SiO_2_, Pro content was reduced in relation to the plants exposed to GLY alone, showing values identical to the CTL.

Regarding GSH [shoots: F (5, 20) = 7.52; *p* < 0.05; roots: F (5, 15) = 10.80; *p* < 0.05], its content was significantly high in shoots of plants from all the treatments when compared to the CTL. In roots, GLY alone provoked a significant increase (100%) in GSH content (Figure 5b,e). However, plants from the co-treatments with Si or nano-SiO_2_ displayed root levels of GSH identical to those found in the CTL plants, especially in GLY + Si treated plants (Figure 5e).

Total AsA levels [shoots: F (5, 19) = 6.71; *p* < 0.05; roots: F (5, 21) = 5.85; *p* < 0.05] are presented in Figure 5c,f. As can be observed, in shoots the total levels of this antioxidant were increased by 34% in GLY− and GLY+ nano-SiO_2_-treated plants, compared to the CTL. Regarding roots, differences in total AsA were only found in plants treated with nano-SiO_2_ alone, with an increase of 39% compared to the CTL.

Concerning the ratio between AsA/DHA [shoots: F (5, 16) = 10.76; *p* < 0.05; roots: F (5, 20) = 22.70; *p* < 0.05], results are compiled in Table 2. Reductions of up to 62% over the CTL were detected in the shoots in all experimental groups. In roots, GLY increased this parameter by 32% in relation to the CTL, this effect being counteracted by the application of Si or nano-SiO_2_, in which AsA/DHA levels were identical to the CTL. Moreover, both forms of Si, when applied alone, led to a reduction of about 50% in AsA/DHA, over the CTL (Table 2).

### 3.5. Enzymatic Antioxidants—Activity of SOD, CAT, APX, GR, and DHAR

As can be observed in Figure 6a, SOD activity was significantly changed in the shoots [F (5, 14) = 3.99; *p* < 0.05], but no differences were recorded between GLY and the CTL. However, plants co-treated with Si, especially nano-SiO_2_, showed increased activity of this enzyme in comparison with plants only exposed to the herbicide. In roots [F (5, 13) = 49.60; *p* < 0.05], GLY significantly inhibited SOD activity by 40% over the CTL (Figure 6d). Once again, plants simultaneously exposed to Si or nano-SiO_2_ and GLY displayed a significantly improved SOD activity, with values even higher (up to 63%) than those found in the CTL roots (Figure 6d). Moreover, the application of nano-SiO_2_ alone reduced the activity of this enzyme.

Concerning CAT and APX, their activity values are shown in Figure 6b,c,e,f. As illustrated, both enzymes showed the same pattern in shoots [CAT: F (5, 18) = 11.76; *p* < 0.05; APX: F (5, 12) = 21.47; *p* < 0.05] and roots [CAT: F (5, 12) = 34.75; *p* < 0.05; APX: F (5, 12) = 67.64; *p* < 0.05] of tomato plants. In general, CAT and APX activity were negatively affected by GLY alone, but plants under the co-treatment with Si or nano-SiO_2_, especially bulk Si, displayed enzyme activity levels identical or even higher than those of the CTL plants. For instance, while CAT and APX activity in roots suffered a decrease of around 0.6-fold induced by GLY, the foliar application of Si enhanced the total activity of both enzymes, with increments up to 3.6- (CAT) and 7-fold (APX) compared to the plants only exposed to the herbicide (Figure 6e,f). As in the case of SOD, the foliar spraying of plants with Si or nano-SiO_2_ seemed to have diminished the activity of these two antioxidant enzymes (Figure 6b,c,e,f).

The quantification of GR and DHAR activity showed distinct patterns in shoots [GR: F (5, 12) = 55.33; *p* < 0.05; DHAR: F (5, 12) = 26.49; *p* < 0.05] and roots [GR: F (5, 13) = 2.94; *p* > 0.05; DHAR: F (5, 13) = 4.41; *p* < 0.05] (Figure 7). Regarding GR, GLY did not affect its activity in the shoots; nevertheless, the foliar spraying of plants to Si or nano-SiO_2_, independently of GLY co-exposure, significantly improved the activity of this antioxidant enzyme, with values even higher than those found in the CTL (Figure 7a). In roots, no changes were recorded (Figure 7c).

Lastly, concerning DHAR, GLY induced a decrease in its activity (ca. 30%) over the control in shoots; however, the application of Si or nano-SiO_2_ alone or in combination with GLY exposure contributed to enhanced activity of this enzyme, with values exceeding those of GLY-treated plants by 37% (Figure 7b). In general, no major changes were found in roots (Figure 7d). 

### 3.6. Bioaccumulation of GLY in Shoots and Roots

As can be observed, GLY was not detected in shoots of any experimental group (Figure 8). In contrast, roots of plants exposed to the herbicide alone displayed GLY levels up to 14.2 µg g^−1^ d.w. However, in response to the foliar application of Si or nano-SiO_2_, GLY bioaccumulation was reduced, with significant decreases of 17 and 13%, respectively. AMPA was not detected in any sample, being below the detection limit.

## 4. Discussions

Recently, our research group, along with other notable studies [12,40,41,42], has provided important findings concerning GLY non-target phytotoxicity in important plant species, such as barley (*Hordeum vulgare* L.) [43], alfafa (*Medicago sativa* L.) [44] and tomato (*Solanum lycopersicum* L.) [10,45]. However, as well as understanding GLY-associated environmental risks, it is also essential to develop new approaches to increase plant tolerance to this herbicide [11]. Therefore, the main goal of the present study was to explore the potential of Si, either in its bulk or nano formulations, to overcome GLY-induced oxidative stress in tomato plants.

### 4.1. GLY-Mediated Inhibition of Plant Growth Is Efficiently Counteracted by the Foliar Application of Si or Nano-SiO_2_

The occurrence of phytotoxic symptoms, along with the inhibition of plant growth performance, is among the most common effects of soil contamination on plants [46,47]. As expected, the exposure of tomato plants to 10 mg kg^−1^ GLY (levels already found in agricultural fields [44]) resulted in a marked decrease in growth traits, with significant reductions in both roots and shoots. These observations are in agreement with our previous report, in which the development of *S. lycopersicum* cv. Micro-Tom was severely hampered by increasing concentrations (10, 20 and 30 mg kg^−1^) of GLY residues in soil [10]. The same pattern has already been observed for other plant species grown in GLY-contaminated media, with negative impacts in both dicot (pea, willow and alfalfa plants) and monocot (barley and rice plants) species [11,12,13,44,48]. The phytotoxic hazards of GLY in plant growth are probably related to its mode-of-action, where amino acid biosynthesis is hampered, thus compromising cell homeostasis and plant growth, but it can also reflect the interference of GLY with mineral nutrition. By acting as a metal chelating agent, GLY may reduce the uptake of different essential nutrients and/or decrease their bioavailability inside plant tissues (reviewed by Gomes et al. [7]). However, research has not reached consensus, with studies reporting contrasting findings on this matter (see review by Mertens et al. [49]), GLY was found to reduce the leaf levels of Ca, Mn, Fe and Mg in a non-resistant soybean genotype [50]. This suggests that it can affect not only root uptake of these elements, but also root-to-shoot transport. Moreover, GLY is known to form complexes with divalent cations (e.g., Ca^2+^. Mg^2+^, Fe^2+^), resulting in nutrient immobilization inside plant tissues [50,51,52,53,54]. Although EPSPS is a chloroplastidial enzyme [55,56], the phytotoxic action of GLY was preferentially observed in roots, where a much sharper decrease in biomass production was observed. Accordingly, bioaccumulation data showed that roots were the preferential organ for GLY storage, with highly limited translocation for the aerial parts. Even when foliar-applied, GLY has been found to spread to different plant organs, particularly accumulating in tissues with a high metabolic activity, such as root and shoot apexes [7]. Thus, it can be suggested that GLY-mediated stress in tomato plants is not strictly related to its specific herbicidal activity, which was designed to target the EPSPS enzyme, but also emerges as the result of GLY secondary effects on plant physiology. Moreover, although GLY was not detected in the aerial organs of tomato plants, and thus seems to not represent a threat to food safety, its risks in terms of food security should not be neglected, since plant growth was majorly impaired.

The potential of Si to enhance plant abiotic stress tolerance is widely recognized, offering protection against several types of environmental stresses, including abiotic (drought, salinity, metals) and biotic (pathogens, virus, herbivore attack) factors [19,57,58]. However, up to now, this is the first study reporting the effective potential of Si to increase GLY tolerance in crops. Upon Si foliar spray, GLY-induced phytotoxic effects were partially or almost completely inhibited (Figure 1 and Figure 2). Not only did plants show a better growth potential at the macroscopic level, the results concerning biomass production and organ elongation further confirmed this trend. Curiously, the only available record exploring the ameliorative features of Si nutrition against herbicide toxicity was recently published [59]. In agreement with our data, Tripathi et al. [59] also observed a positive effect of Si, provided as 10 µM sodium silicate, when 25-day-old rice plants were exposed to butachlor (4 µM). Furthermore, Si-mediated alleviation of GLY phytotoxicity seems also to be related to a reduced herbicide bioaccumulation, since GLY levels decreased in roots of Si and nano-SiO_2_ treated plants. The ability of Si to prevent contaminant accumulation in different plant species has often been reported in the literature [15,18], which is in accordance with our data.

Si-mediated stimulation of plant growth, either under optimal or stressful conditions, can result from its role in maintaining a proper water balance in plants [60,61]. In fact, Si application can modulate the transcript levels of aquaporin-related genes, contributing to better water absorption [62,63] and, consequently, improved nutrient uptake and translocation [64]. As can be observed, the co-treatment of plants with GLY and Si or GLY and nano-SiO_2_ contributed to better plant growth, lowering the observed damage induced by GLY in root length and root and shoot fresh biomass. Despite the lack of studies, this result was an expected outcome given the widely recognized ability of Si, both at bulk and nano forms, to enhance plant stress tolerance, namely to metal(loid)s, whose detoxification pathways are somewhat similar to those of organic xenobiotics [65]. Indeed, Si supplementation (2 mM sodium silicate) to the nutrient solution helped to reduce cadmium (Cd) phytotoxicity in tomato plants [66]. In parallel, when studying the potential of Si to alleviate the effects of nickel oxide nanomaterials (nano-NiO) in *H. vulgare*, Soares et al. [21] reported an increased plant growth performance upon soil amendment with nano-SiO_2_ (3 mg kg^−1^). Equivalent observations were also described for other plant species, treated with Si, exposed to different metals, including Cd [67,68], chromium (Cr) [69,70] and aluminium (Al) [71,72].

Apart from the overall beneficial effects of Si against GLY-induced toxicity in tomato plants, no substantial differences were detected between the two applied forms of Si, though different studies report that nanotechnology-based tools can be more efficient than their bulk counterparts [73,74]. However, and as reported herein, a previous study also conducted with *S. lycopersicum* plants, revealed that Si-mediated salinity tolerance did not differ between bulk (silicate—1 and 2 mM) and nano formulations (nano-SiO_2_—1 and 2 mM) [75]. However, since no solid conclusions can be drawn only based on biometric parameters, analyses then focused on the evaluation of the redox homeostasis of GLY-exposed plants treated, or not, with Si or nano-SiO_2_.

### 4.2. The Foliar Application of Si or Nano-SiO_2_ Reduces GLY-Induced Oxidative Stress, Particularly Stimulating the Enzymes of the Antioxidant Defense System

Although GLY primary target is not related to redox disorders, it is widely accepted that, once in plant cells, GLY is able to disturb the redox homeostasis [7,76]. Our results clearly showed that GLY residues in the soil ended up affecting the overall redox state of tomato plants. In general, data concerning ROS quantification and LP degree evaluation seemed to suggest the occurrence of oxidative stress as a response to GLY exposure, which agrees with our previous study [10]. Indeed, and although LP and H_2_O_2_ have not followed the same trend in roots and shoots, O_2_^•−^ content revealed to be greatly enhanced upon exposure to GLY. Although this ROS is considered to be a moderate reactive radical, since it has a short half-life, is negatively charged and does not have the ability to cross biological membranes, O_2_^•−^ can further give rise to the production of other, more oxidizing, agents, including hydroxyl radical (^•^OH), through the Haber–Weiss reaction, and hydroperoxyl (HO_2_^−^), through protonation, the latter being permeable and highly reactive [77]. Moreover, and considering that EPSPS is located in chloroplasts [55,56], the main source of O_2_^•−^ in plant cells, the burst of this ROS in GLY-exposed plants suggests, once again, that tomato plants failed to prevent the occurrence of oxidative stress. Accordingly, in our former study, a completely altered ultrastructure of chloroplasts towards GLY treatment was observed [45], reinforcing that plastid-mediated changes can be overall indicators of GLY-induced stress in plants.

Although the involvement of Si in alleviating pesticide-induced oxidative damage is somehow unexplored, the widely recognized ability of this element to counteract the toxic effects of ROS on membrane and organelle damage led us to hypothesize that beneficial effects would be recorded upon the application of Si to GLY-exposed plants. Indeed, supporting the results of the biometric assessment, tomato plants grown under GLY exposure but simultaneously treated with Si or nano-SiO_2_ were able to maintaining the redox homeostasis, with lower levels of O_2_^•−^ in both organs and MDA in shoots. One of the ways by which Si is able to increase plant abiotic stress tolerance is by reducing oxidative stress, given its ability to enhance the antioxidant performance [78]. Positive effects of Si, along with nano-SiO_2_, on the prevention of ROS overproduction and membrane damage in stressed plants are quite common in the literature and strongly point towards the potential of this beneficial element for plant stress management [18,19]. Up to now, the potential of Si to overcome pesticide-induced toxicity is limited to a very recent work, conducted by Tripathi et al. [59] in rice plants treated with butachlor. In that study, authors observed that Si’s ameliorative action was linked with an improvement in nutrient uptake, maintenance of the photosynthetic potential and prevention of oxidative damage, through an upregulation of the AsA-GSH cycle.

GLY exposure resulted in differential responses between the enzymatic and non-enzymatic antioxidant components. In general, an overaccumulation of Pro, AsA and GSH took place in both organs, but an overall downregulation of the enzymatic mechanisms was perceived. Often, plant responses to abiotic stress factors, including xenobiotic exposure, result in differentially activated/inhibited players. Pro stimulation as a consequence of GLY is a common response of different plant species [10,11,12,42,44]. Accordingly, one of the most recurrent symptoms of GLY at the cellular level is the overaccumulation of this proteinogenic amino acid, whose involvement in stress tolerance has been recurrently highlighted [79]. The observed rises in Pro content against GLY underpin that plant cells are able to sense and respond to GLY intracellularly, attempting to limit its toxicity. However, an upsurge of this antioxidant is not always synonymous with enhanced tolerance, but rather a stress signal. Curiously, the magnitude of these increases should be carefully interpreted. Interested in unravelling the role of Pro accumulation against salt stress in 30 wheat (*Zea mays* L.) cultivars, Poustini et al. [80] found that all of them increased Pro levels, but the most sensitive were the ones reporting the highest increases. Hence, Pro is believed to act not always as a tolerance mechanism, but also as one of the earliest metabolic signals upon exposure to stress, capable of inducing other regulatory networks [81]. In accordance with this hypothesis, the prevention of a GLY-mediated burst of O_2_^•−^ and LP, as well as H_2_O_2_ in roots, by both sources of Si was not accompanied by a great increase in Pro accumulation as was observed in plants only exposed to GLY. Thus, although several reports state that Si application can boost Pro levels to increase the antioxidant efficiency [82], the findings reported herein confirm the hypothetical role of Pro as a stress signal, rather than as a tolerance mechanism against herbicide exposure. Similarly, Tripathi et al. [59], Spormann et al. [11] and Sousa et al. [83] found that pesticide- and Zn-mediated increases in Pro levels were efficiently counteracted by the application of Si, salicylic acid, and brassinosteroids (24-epibrassinolide), respectively. Thus, it appears that Si and nano-SiO_2_’s ameliorative effects on ROS homeostasis and LP in shoots are not related to the action of Pro, but rather to other antioxidant components. 

In the current study, the activity of the main ROS-detoxifying enzymes was severely repressed by GLY. Especially in shoots, where MDA and O_2_^•−^ levels rose, the explanation of this inhibition can be ascribed to the effects of oxidative stress itself on enzyme activity. Indeed, it is known that some antioxidant enzymes are particularly sensitive to oxidation, resulting in a reduced catalytic capacity [84]. Moreover, the significant decrease in protein-bond thiols (-SH groups) (data not shown) in shoots further supports that antioxidant enzymes failed to prevent the negative effects of GLY on tomato plants. In contrast to plants only exposed to GLY, O_2_^•−^ and H_2_O_2_-neutralizing antioxidant enzymes were found to be generally increased upon the co-treatment with Si, especially when this element was provided in its bulk form. SOD is usually considered as the first enzymatic line of defence against oxidative stress, being capable of neutralizing the toxic effects of O_2_^•−^. While SOD activity was restored back to CTL values in shoots and stimulated in roots, thus explaining the generalized reduction of O_2_^•−^ in both organs, CAT and APX performance were upregulated, particularly when compared to plants exposed only to GLY, helping to maintain the controlled levels of H_2_O_2_. In agreement with APX activity, the other studied AsA-GSH cycle-related enzymes were also enhanced in shoots in response to the co-treatments, reinforcing Si-mediated activation of the antioxidant system, particularly the enzymatic one. Integrating these responses as a whole, it can be strongly suggested that Si promoted a positive redox balance that allows protein stability and redox state. In fact, in plants exposed to GLY, but simultaneously treated with Si or nano-SiO_2_, protein thiols were increased in shoots (data not shown), despite GLY’s negative effect on this parameter. In a previous study, the involvement of Si in stimulating the thiol-based redox network was also suggested [21,85]. Although studies exploring ways to increase GLY tolerance of non-target plants are numerous, some of them highlight the importance of the enzymatic antioxidant system in this response. When studying the potential of salicylic acid (100 µM) and nitric oxide (NO; 250 µM) to ameliorate GLY-induced oxidative stress in barley and pea plants, respectively, Spormann et al. [11] and Singh et al. [13] reported that different antioxidant enzymes, including CAT, SOD and APX, were much more efficient under the co-treatments, limiting GLY’s harmful effects on ROS overproduction. In opposition, by unravelling the modulation of GLY toxicity by the supplementation of phosphate (PO_4_^3−^), a general inhibition of the activity of the main antioxidant enzymes (e.g., SOD, CAT, APX) was found in GLY-treated *Hydrocharis dubia* (Bl.) Backer upon P supplementation [86]. 

Considered as the main antioxidant buffers of plant cells, the reduced state of AsA and GSH determines the whole redox balance of the cell [87]. As shown, GSH and AsA accumulation was stimulated in response to the presence of the herbicide. Following the herbicide detoxification system, it is not a surprise that GSH levels rose in shoots and roots of GLY-treated plants. Indeed, the conjugation of herbicides with GSH, via glutathione-S-transferase (GST; EC 2.5.1.18) is one of the most common reactions occurring during xenobiotic detoxification [88]. This increase was higher in roots than in shoots, suggesting that the main organ responsible for GLY degradation is the root system, in an attempt to limit its translocation to the aerial parts. Acting in tandem with GSH, through the involvement of the AsA-GSH cycle, ascorbate forms (AsA and DHA) were also significantly altered by GLY, generally showing increased levels. Thus, recalling that H_2_O_2_ did not differ between CTL and GLY plants, it can be hypothesized that non-enzymatic mechanisms, rather than enzymatic ones, have dealt with the excess of this ROS, preventing its accumulation. Regarding this matter, both AsA and GSH are capable of directly eliminating this ROS [79].

Despite this marked stimulation of the antioxidant metabolites towards GLY exposure, upon co-treatment with both forms of Si, their content did not substantially differ relative to plants exposed only to the herbicide. In fact, regardless of the overall upregulation of the AsA-GSH cycle enzymes, the levels of AsA and GSH were somehow identical to those of GLY-treated plants. However, despite the total levels of both antioxidant being unchanged, their biological relevance is distinct, given the recorded effects on enzyme kinetics and ROS content. Indeed, while total AsA increased in GLY-treated plants, the observed decreases in AsA/DHA ratio suggest different regulatory phenomena: while in GLY single treatments, this lower ratio was not accompanied by a stimulation of APX activity, but rather a rise in MDA and O_2_^•−^, the same was not verified in response to the co-treatments. With the effect being, under the joint action of Si and GLY, APX and DHAR activities were increased in shoots, aligned with a decrease in the same oxidative stress markers. Concerning GSH, although some reports suggest that both Si and nano-SiO_2_ are able to boost its levels as a protective effect against the toxic action of ROS, an identical (in the shoots) or even lower (in the roots) content of this antioxidant was found in response to the simultaneous application of Si and GLY. Bearing in mind the proved role of GSH as a xenobiotic conjugating agent [89], these results agree with the hypothesis that both forms of Si were efficient at limiting GLY uptake. From a similar way, and knowing that metals and xenobiotics share a common process of detoxification, including sensing, uptake and storage mechanisms [90], this finding is not surprising. Thus, while in shoots GSH is probably being recruited for antioxidant defence, in roots, its role is probably linked to GLY detoxification, especially in plants solely exposed to the herbicide. 

In terms of efficiency, both sources of Si exhibited interesting potential to increase tomato plants’ tolerance to GLY, sharing mechanisms of action and enhancing similar antioxidant players, especially at the enzymatic level. Since Si nanomaterials display a greater surface area and a higher reactivity, some reports suggest that nano-based solutions are often more appealing than their bulk counterparts [73,74,91]. However, other studies found no substantial differences [75] or, quite the opposite, have indicated that bulk sources of Si are a better option for boosting plant abiotic stress tolerance [82]. Yet, in order to fully unravel the potential of nanotechnology-based solutions, additional studies are needed, namely, to test other types of Si NMs, as well as several concentrations and modes of application. NMs physical-chemical features (e.g., size, ionic charge and morphology), by modulating the formation of aggregates, can possibly limit their biological effects [92,93]. Therefore, one cannot exclude that, at the concentration tested, nano-SiO_2_ could have formed aggregates, lowering its ameliorative potential, and bringing it closer to that found for conventional Si. Nevertheless, from a wide perspective, and although in some cases bulk Si led to a more proactive performance of the antioxidant enzymes, such as SOD, CAT and APX, our data suggest that no major aspects differed from the two applied sources of Si, with both of them showing promising effects for increasing crop resilience to GLY residues.

## 5. Conclusions

Overall, our results highlight the urgency for additional studies attempting in preventing GLY-associated risks to non-target plants, especially crops, since residual levels of this herbicide are still capable of inducing phytotoxicity and impairing plant growth. Moreover, being the first report on the effect of Si against GLY toxicity, our data strongly suggest that Si or nano-SiO_2_ can be good candidates for plant stress management approaches, especially from an eco-friendly and sustainable perspective. By applying different complementary approaches, the main mechanisms behind Si-mediated protection towards GLY were unravelled, namely, limitation of GLY uptake and the efficiency of the enzymatic antioxidant system being the main factors behind the higher tolerance of plants sprayed with Si, either as bulk or nano formulations.

## Figures and Tables

**Figure 1 antioxidants-10-01320-f001:**
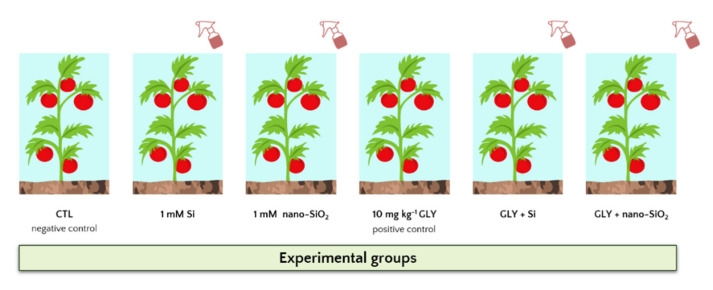
Graphical representation of the experimental design, detailing the main treatments.

**Figure 2 antioxidants-10-01320-f002:**
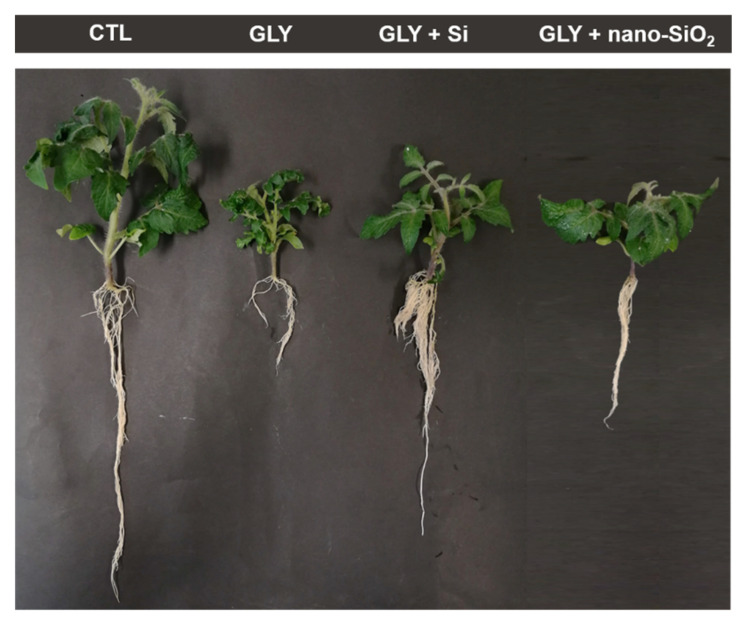
*S. lycopersicum* plants after four weeks of growth (CTL—control plants; GLY—plants exposed to GLY alone; GLY + Si—plants exposed to GLY and treated with Si; GLY + nano-SiO_2_—plants exposed to GLY and treated with nano-SiO_2_).

**Figure 3 antioxidants-10-01320-f003:**
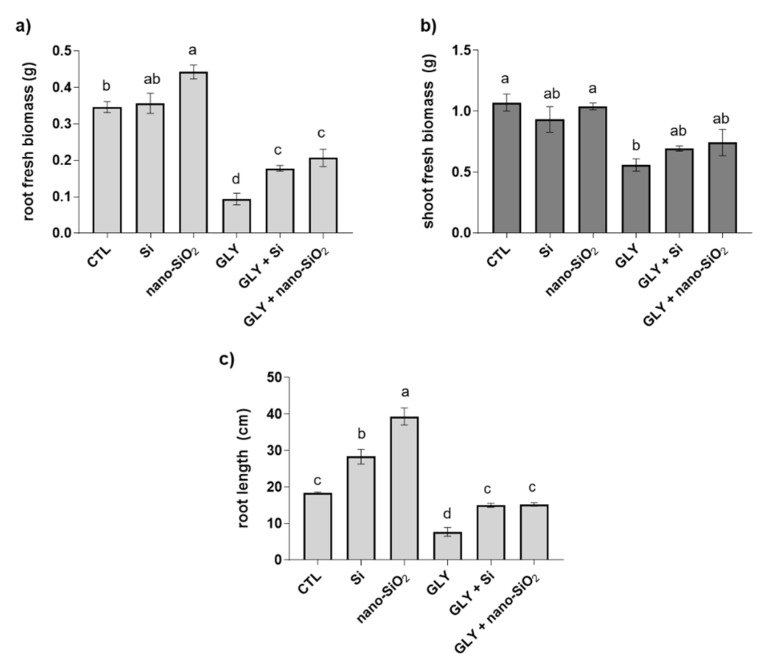
Biometric parameters of *S. lycopersicum* plants after four weeks of growth [CTL—control plants; GLY—plants exposed to GLY alone; GLY + Si—plants exposed to GLY and treated with Si and GLY + nano-SiO_2_—plants exposed to GLY and treated with nano-SiO_2_]. (**a**) Root fresh biomass; (**b**) shoot fresh biomass and (**c**) root length. Data presented are mean ± STDEV (*n* ≥ 3). Different letters (a–d) above bars indicate significant statistical differences between treatments (Tukey: *p* ≤ 0.05).

**Figure 4 antioxidants-10-01320-f004:**
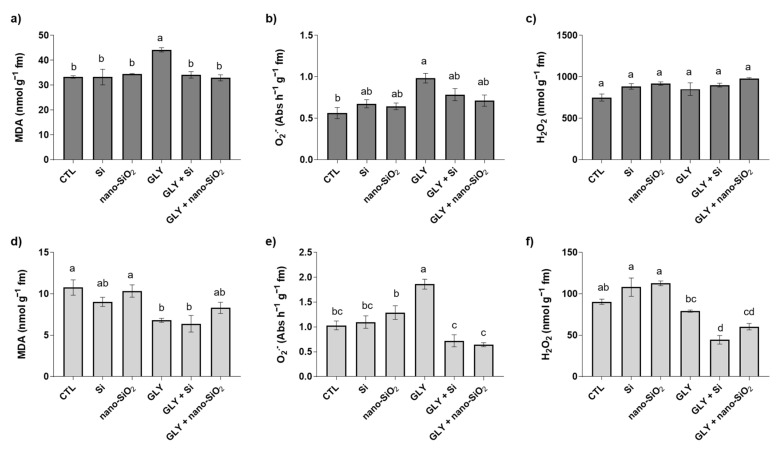
Oxidative stress markers of *S. lycopersicum* plants after 4 weeks of growth. (**a**,**d**) malondialdehyde (MDA); (**b**,**e**) superoxide anion (O_2_^•−^) and (**c**,**f**) hydrogen peroxide (H_2_O_2_). Dark and light bars represent shoots and roots, respectively. Data presented are mean ± STDEV (*n* ≥ 3). Different letters (a–d) above bars indicate significant statistical differences between treatments (Tukey: *p* ≤ 0.05).

**Figure 5 antioxidants-10-01320-f005:**
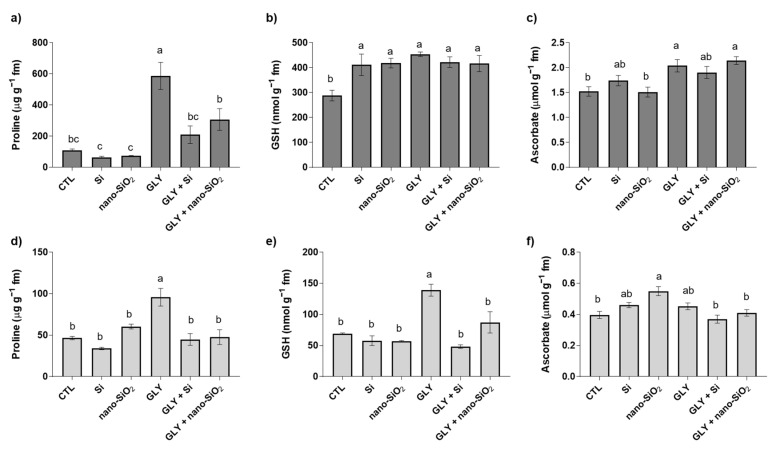
Levels of the main antioxidant metabolites of *S. lycopersicum* plants after 4 weeks of growth. (**a**,**d**) proline (Pro); (**b**,**e**) glutathione (GSH); (**c**,**f**) total ascorbate. Dark and light bars represent shoots and roots, respectively. Data presented are mean ± STDEV (*n* ≥ 3). Different letters (a–c) above bars indicate significant statistical differences between treatments (Tukey: *p* ≤ 0.05).

**Figure 6 antioxidants-10-01320-f006:**
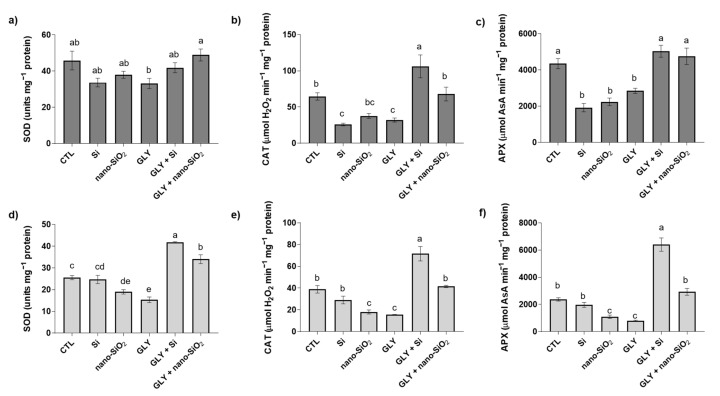
Total activity of superoxide dismutase (SOD; **a**,**d**) and catalase (CAT; **b**,**e**) and ascorbate peroxidase (APX; **c**,**f**) of *S. lycopersicum* plants after 4 weeks of growth. Dark and light bars represent shoots and roots, respectively. Data presented are mean ± STDEV (*n* ≥ 3). Different letters (a–e) above bars indicate significant statistical differences between treatments (Tukey: *p* ≤ 0.05).

**Figure 7 antioxidants-10-01320-f007:**
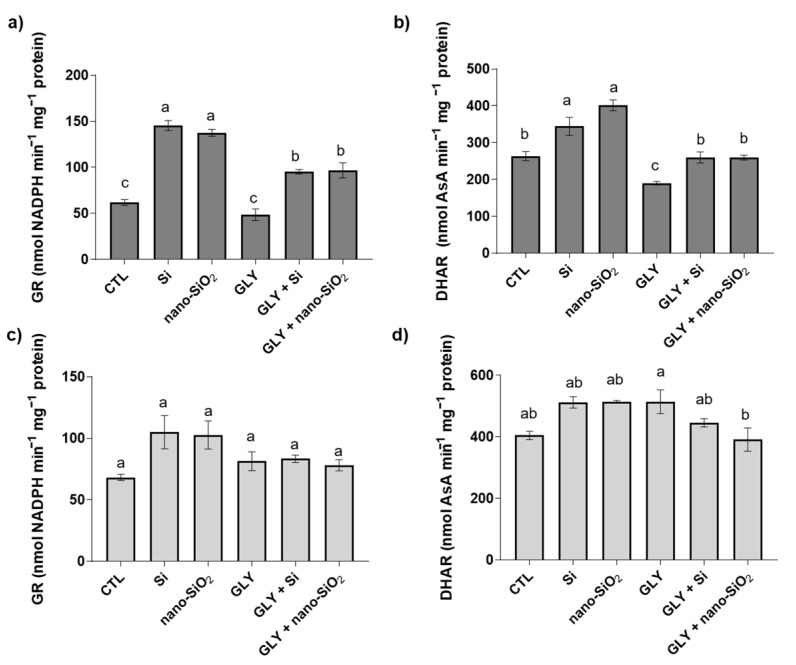
Total activity of glutathione reductase (GR; **a**,**c**); dehydroascorbate reductase (DHAR; **b**,**d**) of *S. lycopersicum* plants after 4 weeks of growth. Dark and light bars represent shoots and roots, respectively. Data presented are mean ± STDEV (*n* ≥ 3). Different letters (a–c) above bars indicate significant statistical differences between treatments (Tukey: *p* ≤ 0.05).

**Figure 8 antioxidants-10-01320-f008:**
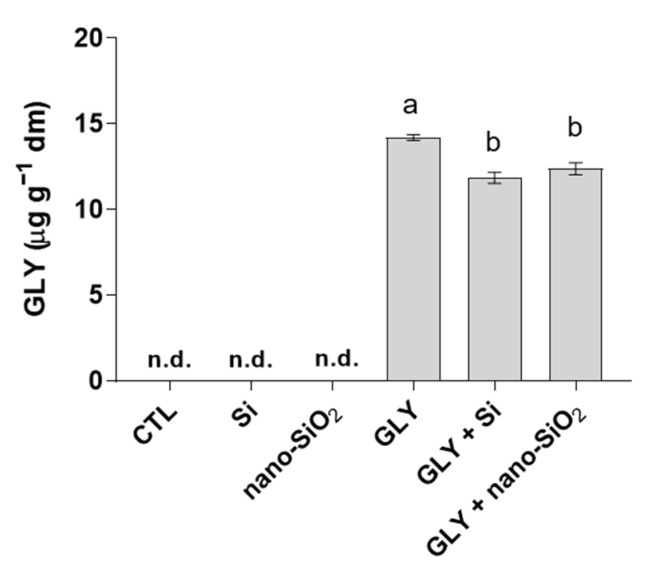
GLY levels in roots of *S. lycopersicum* plants after 4 weeks of growth. n.d.: non-detected, which means below the detection limit. Data presented are mean ± STDEV (*n* ≥ 3). Different letters (a, b) above bars indicate significant statistical differences between treatments (Tukey: *p* ≤ 0.05).

**Table 1 antioxidants-10-01320-t001:** Multiple reaction monitoring (MRM) transitions, cone voltages and collision energies for each used compound.

Compound	Precursor Ion (*m*/*z*)	Product Ion (*m*/*z*)	Cone Voltage (V)	Collision Energy (V)
GLY-FMOC	392.2	Q:88.0	20	20
		q:170.0	20	10
1,2-^13^C2, ^15^N GLY-FMOC	395.2	91.0	20	20
AMPA-FMOC	334.0	Q:112.1	20	15
		q:179.1	20	20
^13^C,^15^N-AMPA	336.0	114.1	20	15

Q: quantification transition; q: confirmation transition.

**Table 2 antioxidants-10-01320-t002:** Levels of total AsA (µmol g^−1^ fm), along with its reduced and oxidized forms (dehydroascorbate—DHA), of *S. lycopersicum* plants after 4 weeks of growth. Data presented are mean ± STDEV (*n* ≥ 3). Different letters (a–c) indicate significant statistical differences between treatments (Tukey: *p* ≤ 0.05).

Organ	Treatment	Total AsA	AsA/Total AsA	DHA/Total AsA	AsA/DHA
Shoots	CTL	1.52 ± 0.09 b	0.72 ± 0.05 a	0.28 ± 0.05 a	3.44 ± 0.66 a
	Si	1.74 ± 0.10 ab	0.59 ± 0.06 a	0.41 ± 0.06 a	1.55 ± 0.40 b
	Nano-SiO_2_	1.51 ± 0.10 b	0.56 ± 0.05 a	0.44 ± 0.05 a	1.31 ± 0.27 b
	GLY	2.04 ± 0.12 a	0.56 ± 0.03 a	0.44 ± 0.03 a	1.47 ± 0.22 b
	GLY + Si	1.90 ± 0.12 ab	0.60 ± 0.05 a	0.40 ± 0.05 a	1.48 ± 0.30 b
	GLY + nano-SiO_2_	2.14 ± 0.08 a	0.63 ± 0.02 a	0.37 ± 0.02 a	1.65 ± 0.19 b
Roots	CTL	0.40 ± 0.02 b	0.41 ± 0.03 bc	0.59 ± 0.03 ab	0.77 ± 0.05 b
	Si	0.46 ± 0.02 ab	0.33 ± 0.04 c	0.67 ± 0.04 a	0.42 ± 0.06 c
	Nano-SiO_2_	0.55 ± 0.03 a	0.33 ± 0.03 bc	0.67 ± 0.03 a	0.38 ± 0.04 c
	GLY	0.45 ± 0.02 ab	0.50 ± 0.02 a	0.50 ± 0.02 bc	1.02 ± 0.04 a
	GLY + Si	0.37 ± 0.03 b	0.45 ± 0.01 ab	0.49 ± 0.02 c	0.79 ± 0.03 b
	GLY + nano-SiO_2_	0.41 ± 0.02 b	0.44 ± 0.01 ab	0.56 ± 0.01 abc	0.80 ± 0.05 b

## Data Availability

The data presented in this study are available in this manuscript.

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
