# Peer review of "Silicon Improves the Redox Homeostasis to Alleviate Glyphosate Toxicity in Tomato Plants—Are Nanomaterials Relevant?"

_antioxidants, 2021, doi:10.3390/antiox10081320_

Round 1

Reviewer 1 Report

The manuscript reports a straightforward study on the protective effects of silicon on glyphosate-stressed tomato plants. The experiments were done correctly with a number of replicates and a reasonable number of samples, and the parameters chosen for determinations are well based and adequately described. The discussion keeps well to the point and addresses various physiological parameters to explain the positive effects of silicon application. The conclusions drawn are very well supported by the data even though the actual metabolic reasons for the effects could not be pointed out. The English language is excellent.

Minor corrections:

Line 59. chorismate, not chorismite

Line 196. Please mention how the reaction mixture was illuminated for the SOD assay based on photoreduction of NBT.

Figure 4. Why are figures 4b and 4e labelled as Abs g-1, when in the text it written that the SOD units were calculated? (In Figure 6 the SOD units are shown.)

Author Response

Dear Reviewer,

Please see the attached file with our detailed response.

Best regards,

Reviewer 2 Report

There is a lot of literature on glyphosate. There is much less literature on Si. Combining both, as done here, is unique and original. The article shows the negative effects of glyphosate on tomato plants and how these can be counteracted by the supply of Si in different formulations. This is well documented by analyses of responses at a large scope of different functional levels in the plants, namely biometric parameters, oxidative stress markers, AOX metabolites, oxidative stress enzymes. It is of basic interest for redox homeostsis and has practical implications for agriculture. The presentation is fine, well organized and easy to follow, well written text easy to read. Congratulations on a valuable contribution.

Studying the article I came across a number of minor things which are worth considering in a minor revision, as follows:

L188 a letter is missing: bipyridyl

L282 rearrange wording: … this effect being significantly counteracted …

L288 could O2.- be put to the ordinate to make it more clear?

L315 1.0-fold is not a rise but just equal

L320 “similar in shoots of plants” are you sure? It is lower in CTL, there is a letter b on top, versus a in the other cases

L333 rearrange wording: … this effect being counteracted …

Table 2 for total AsA which is not a ratio, the unit has to be given in the table, according to methods-section µmol g-1 fm

L364 diminished

L439, 619 ref.-numbers not bold

L491 rearrange wording: … the latter being permeable 

Author Response

Dear Reviewer,

Please see the attached file with our detailed responses.

Best regards,
